# Development of a Clinical–Biological Model to Assess Tumor Progression in Metastatic Pancreatic Cancer: Post Hoc Analysis of the PRODIGE4/ACCORD11 Trial

**DOI:** 10.3390/cancers14205068

**Published:** 2022-10-16

**Authors:** Julie Egea, Julia Salleron, Sophie Gourgou, Ahmet Ayav, Valérie Laurent, Béata Juzyna, Alexandre Harlé, Thierry Conroy, Aurélien Lambert

**Affiliations:** 1Department of Medical Oncology, Institut de Cancérologie de Lorraine, 54500 Vandœuvre-lès-Nancy, France; 2Biostatistic Unit, Institut de Cancérologie de Lorraine, 54500 Vandœuvre-lès-Nancy, France; 3Biometrics Unit, Institut régional du Cancer Montpellier Val d’Aurelle, Université de Montpellier, 34298 Montpellier, France; 4Department of Gastrointestinal Surgery, Centre Hospitalier Universitaire de Nancy, 54500 Vandœuvre-lès-Nancy, France; 5Department of Radiology, Centre Hospitalier Universitaire de Nancy, 54500 Vandœuvre-lès-Nancy, France; 6UNICANCER Research and Development Team, 75654 Paris, France; 7Department of Biopathology, Institut de Cancérologie de Lorraine, 54500 Vandœuvre-lès-Nancy, France

**Keywords:** pancreatic cancer, tumor progression, quality of life, CA 19-9, patient-reported outcomes

## Abstract

**Simple Summary:**

There is no valid consensus regarding follow-up for pancreatic cancer in terms of recommended investigations in routine practice. By analogy with other pathologies, cross-sectional imaging is often performed every 3 months. In addition, the low number of available chemotherapy regimens does not allow for the envisioning of a great improvement in the life expectancy of the patients, even in cases with fast detection of tumor progression. Our study, therefore, seeks to find a complementary approach to conventional imaging examinations to detect tumor progression more reliably in patients treated for pancreatic cancer, allowing early treatment and thus possibly a better prognosis. This approach consists of a combination of image analysis, biological parameters, and patient self-assessment.

**Abstract:**

Background: The follow-up of pancreatic cancer (PC) is based on computed tomography (CT) assessment; however, there is no consensus on the use of clinical and biological criteria in tumor progression. We aimed to establish a clinical–biological model to highlight the progression of metastatic PC during first-line treatment. Methods: The patients treated with first-line chemotherapy in the phase 2/3 PRODIGE4/ACCORD11 clinical trial were evaluated retrospectively. Clinical and biological markers were evaluated at the time of CT scans and during treatment to determine tumor progression. Results: In total, 196 patients were analyzed, with 355 available tumor assessments. The clinical and biological factors associated with tumor progression in multivariate analysis included gemcitabine, global health status ≤ 33 (OR = 3.38, 95%CI [1.15; 9.91], *p* = 0.028), quality of life score between 34 and 66 (OR = 2.65, 95%CI [1.06; 6.59], *p* = 0.037), carcinoembryonic antigen (CEA) ≥ 3 times the standard value without any increase in the CEA level from inclusion (OR = 2.22, 95%CI [1.01; 4.89], *p* = 0.048) and with an increase in the CEA level from inclusion (OR = 6.56, 95%CI [2.73; 15.78], *p* < 0.001), and an increase in the carbohydrate antigen 19-9 level from inclusion (OR = 2.59, 95%CI [1.25; 5.36], *p* = 0.016). Conclusions: The self-assessment of patients’ general health status alongside tumor markers is an interesting approach to the diagnosis of the tumor progression of metastatic pancreatic cancer patients during first-line treatment.

## 1. Introduction

Pancreatic cancer is the third leading cause of cancer-related deaths in the United States [1]. Its incidence has increased continuously over the last 30 years. It is projected to become the second most common cause of cancer-related deaths in the United States by 2040 [2]. Its prognosis remains very poor, with a 5-year survival rate of 3% for metastatic cancers [1], despite the changes in first-line therapeutic practices with the advent of FOLFIRINOX [3] and the combination of nab-paclitaxel and gemcitabine [4]. More than 80% of patients are diagnosed at a locally advanced or metastatic stage; therefore, their management remains a major challenge for the future.

Computed tomography (CT) scans with Response Evaluation Criteria in Solid Tumors (RECIST) are the standard for assessing responses to treatment in most cancers, including locally advanced or metastatic pancreatic cancer (mPC) [5,6]. However, the ideal timeframe for imaging surveillance remains under discussion [6].

Recently, the first international consensus defined the appropriate mandatory baseline and prognostic data to be used in future pancreatic cancer clinical trials [7], using the data from 39 randomized trials with a total of 15,863 patients. Several parameters can be used to assess cancer progression, including carbohydrate antigen 19-9 (CA19-9) or carcinoembryonic antigen (CEA) [8,9,10]. However, clinical follow-up is mainly based on the Eastern Cooperative Oncology Group Performance Status (ECOG PS) and the monitoring of symptoms such as pain and weight loss, even though these criteria are subjective and their role in prognosis is not well-established [11]. The quality of life assessment seems to be a promising indicator in patient follow-up [12,13] and has well-established prognostic value in mPC [14].

Since no predictive markers are available that can determine progression on their own, we aimed to use all the available tumor evaluations from the database of the phase 2/3 PRODIGE4/ACCORD11 clinical trial to establish a clinical–biological model capable of diagnosing mPC progression (as objectively determined by CT scans) in first-line treatment settings. The challenge consisted of developing a clinical–biological model to diagnose progression at one tumor evaluation by considering only the parameters available at inclusion and at this tumor evaluation regardless of its timing.

## 2. Materials and Methods

### 2.1. Patients

The analyzed patients were selected from the multicenter and randomized phase 2/3 PRODIGE4/ACCORD11 clinical trial, sponsored by UNICANCER and approved by the Unicancer Gastrointestinal Group, involving 342 patients and comparing FOLFIRINOX with gemcitabine as the first-line treatment for mPC. The inclusion procedure took place between January 2005 and October 2009, and the final analysis was conducted on 16 April 2010. The patients were over 18 years of age, had histologically proven and measurable mPC, had not previously been treated with chemotherapy, had a performance status (PS) of 0 or 1, and had to have hematological, hepatic (bilirubin ≤ 1.5 times the upper standard), and renal functions adequate for chemotherapy administration. The patients over 76 years of age, previously treated with radiation therapy, with brain metastases, a history of other cancers, heart disease or chronic diarrhea, and histology other than exocrine pancreatic adenocarcinoma were excluded. For the present study, we selected those patients who had been evaluated by at least one CT scan during follow-up and who had available tumor marker data (CA19-9 and CEA) from the time of the CT scan.

This study was declared to the French National Commission on Informatics and Liberties (CNIL) and received prior authorization from the study sponsor, Unicancer.

### 2.2. Anthropometric and Clinical Parameters

A literature review identified the known prognostic factors for overall survival in first-line mPC [7]. The clinical, tumor, and biological criteria were collected at inclusion, during visits for each chemotherapy cycle, and during tumor evaluations. At the time of inclusion, the clinical criteria were as follows: sex, age, the type of chemotherapy received, ECOG PS, weight, and body mass index (normalized weight squared, BMI). The tumor criteria were as follows: tumor location and the presence of hepatic, peritoneal, pulmonary, and other metastases. The biological criteria were CA19-9, CEA, and albumin levels, which were standardized to the standard of each laboratory.

The clinical data collected during tumor evaluation were as follows: ECOG PS, weight loss since inclusion, and BMI. The patient quality of life data were measured using version 3 of the Quality of Life Core Questionnaire (QLQ-C30) of the European Organization for Research and Treatment of Cancer [15] to evaluate the quality of life in the previous seven days. In our study, the questionnaire was analyzed in accordance with the EORTC guidelines [16]. The analysis focused on the scales that are generally most affected in patients with pancreatic cancer: the global health status and the quality of life scale, as well as other scales, namely fatigue, pain, physical functioning, emotional functioning, role functioning, cognitive functioning, social functioning, and financial difficulties. The most frequent symptoms reported in the questionnaire were dyspnea, insomnia, loss of appetite, constipation, and diarrhea. We used the thresholds defined in the initial quality of life analysis of the PRODIGE4/ACCORD11 study [3,14]. These thresholds were ≤33, 34–66, and ≥67 if enrolment was sufficient (at least 30 assessments for each class); otherwise, they were grouped into two categories. The collected biological data included tumor marker levels (CA19-9 and CEA). They were normalized to the standards of each laboratory. The variation since inclusion was also considered. The data on liver markers, such as total bilirubin, transaminases, and gamma-glutamyl transferase, were also collected and expressed as higher or lower than the standard of each laboratory.

The data on toxicities during treatment were also collected based on the National Cancer Institute Common Terminology Criteria for Adverse Events, version 3.0 [17]. All toxicities were recorded at the follow-up visits for each chemotherapy cycle every 14 days. For each patient, we considered the highest toxicities that occurred between each tumor evaluation. The occurrence of at least one hematological toxicity of grade 2 or grade ≥ 3 during the two months before each tumor evaluation was analyzed. The occurrence of hematological toxicity with grade 2 or grade ≥ 3 could require a treatment delay until a return to grade 1 for the first episode, dose reduction for the second episode, and discontinuation of treatment for the third episode. The occurrence of at least one non-hematological toxicity (other toxicities) of any grade during the two months before tumor evaluation was also analyzed.

Tumor progression was monitored using CT scans every two months, with three successive CT scans from the initiation of treatment for the first six months of treatment. The progression was defined according to the RECIST criteria (version 1.0) [5].

### 2.3. Statistical Analysis

Qualitative parameters were described as frequencies and percentages. The normality of quantitative parameters was investigated using the Shapiro–Wilks test, and these parameters are described as means with standard deviations or as medians and interquartile ranges accordingly.

The relationship between the occurrence of tumor progression and the clinical/biological characteristics recorded at the inclusion or tumor evaluations and the toxicity recorded at the tumor evaluations was investigated by using generalized linear mixed models with a logit link to include both fixed and random effects. The statistical unit was the tumor evaluation. The dependent parameter was tumor progression. In bivariate analysis, the fixed effects were the treatment arm and each independent parameter, i.e., clinical, biological, and toxicity characteristics at each tumor evaluation. The random effect was the patient, to take into account the correlation between the successive tumor evaluations of one patient. The structure of the G-side random effect was chosen by investigating various structures during bivariate analyses based on the ratio of the generalized chi-square statistic: the variability in the data was considered to be properly modeled with no residual overdispersion when its degree of freedom was close to 1. Finally, the compound symmetry structure was chosen. Residual diagnostics were performed by investigating the conditional studentized residuals. All the values of CA19-9 and CEA were normalized to the standard of each laboratory. These normalized values at tumor evaluations were dichotomized according to the median of the normalized value at inclusion. The parameters with a *p*-value less than 0.1 in bivariate analyses adjusted on the treatment arm were selected for the multivariate model. The interaction terms of CEA or CA19-9 parameters were investigated and considered for multivariate analysis. The final multivariate model was obtained using backward selection. The results are presented as adjusted odds ratios and 95% confidence intervals. The significance level was set at *p* < 0.05. Statistical analyses were performed using the SAS software (version 9.4; SAS Institute Inc., Cary, NC, USA).

## 3. Results

Among the 342 patients enrolled in the PRODIGE4/ACCORD11 clinical trial, 196 were evaluated by at least one CT scan during follow-up and had tumor marker data available at the time of the CT scan, corresponding to 355 tumor evaluations (Figure 1). Of the 355 tumor evaluations, 92 progressions were diagnosed by CT scans. The 196 included patients had better overall and progression-free survival than the 146 patients who were not selected (see Appendix A).

Table 1 describes the patient characteristics at inclusion. The mean age was 58.8 years ± 8.8. Seventeen patients (8.7%) were obese (BMI > 30 kg/m^2^). All the patients had metastatic cancer from the beginning, with hepatic metastases in 87.8% of the cases (*n* = 172). CA19-9 levels were higher than the standard value in 87.9% of the patients (*n* = 167, 6 missing) and CEA in 65.1% of the patients (*n* = 125, 4 missing).

Table 2 describes the characteristics of the patients included in the 355 tumor evaluations. In terms of clinical characteristics, the ECOG PS remained at 0 or 1 in 83.6% (*n* = 240, 68 missing) of tumor evaluations. The proportion of tumor evaluations in which the patients exhibited weight loss ≥ 5% was 37.5% (*n* = 133). In over 50% of tumor evaluations, the patients had CA19-9 and CEA levels higher than standard values; CA19-9 had increased since inclusion in 25.3% (*n* = 86, 15 missing) and CEA in 39.4% (*n* = 138, 5 missing) of tumor evaluations. The proportion of tumor evaluations in which the patients had CA19-9 levels greater than 30 times the standard value and CEA levels greater than 3 times the standard value was 43.4% (*n* = 154) and 30.2% (*n* = 104, 11 missing), respectively. In 18.8% of tumor evaluations, the patients reported having quite a bit/very much pain during the past week, and in 51.5% of tumor evaluations, the patients reported an overall health score of >4.

Table 3 presents an analysis of the clinical and biological factors recorded at inclusion or at tumor evaluations that were associated with tumor progression at these tumor evaluations, regardless of their timing.

Table 4 presents an analysis of the quality of life score with the thresholds defined in the PRODIGE4/ACCORD11 quality of life study [14]. These thresholds were ≤ 33, 34–66, and ≥67 if the enrolment was sufficient (at least 30 assessments for each class); otherwise, they were grouped into two categories.

The multivariate analysis found four criteria recorded at tumor evaluations for the diagnosis of tumor progression during those evaluations: the global health status ≤33 (OR = 3.38, 95%CI [1.15; 9.91], *p* = 0.028); the quality of life score between 34 and 66 (OR = 2.65, 95%CI [1.06; 6.59], *p* = 0.037); CEA level ≥ threefold the standard value without any increase in the CEA level from inclusion (OR = 2.22, 95%CI [1.01; 4.89]); CEA level ≥ threefold the standard value with an increase in the CEA level from inclusion (OR = 6.56, 95%CI [2.73; 15.78]); and an increase in the CA19-9 level from inclusion (OR = 2.59, 95%CI [1.25; 5.36]) (Table 5). The area under the curve (AUC) of this multivariate model was 0.78, whereas that for the treatment arm alone was 0.67.

## 4. Discussion

Our study highlights a new approach for diagnosing disease progression based on treatment type, the self-assessment of patients’ overall health status, and CA19-9 levels. This could allow for the earlier diagnosis of tumor progression, compared with CT scan assessment alone, in those patients with mPC during first-line therapy. The discriminating power of our tumor progression diagnosis model is nevertheless too weak to confirm the progression diagnosis with certainty. However, we identified the components necessary for the validation of a clinical–biological score that could enable the diagnosis of tumor progression. In addition, the analyzed patients were not representative of all the patients included in the phase 2/3 PRODIGE4/ACCORD11 clinical trial, as the patients in our study had better survival outcomes. This is due to our selection of those patients who had been evaluated with at least one CT scan with available tumor markers, thus excluding de facto patients with progression before the first CT evaluation. However, our study highlights major elements for patient follow-up and underlines that the development of an efficient tool would allow for an earlier diagnosis of tumor progression than is currently possible with CT scans. This would also allow for the early detection of those patients who respond to treatment, meaning for whom reassessment by CT scans could be delayed. Indeed, for those patients with progression, two months between scans is often too long, given the low long-term survival rate of patients with mPC [6]. On the contrary, identifying patients responding to treatment could reduce the number of scans required and thus the emotional and financial costs they can generate.

Our study revealed that the patient’s self-assessment of poor overall health is intimately linked to tumor progression. It should be noted that we did not have any specific quality of life data on those patients who have had previous surgery; there were 15 and 10 patients in the FOLFIRINOX and gemcitabine arms, respectively, who underwent surgery before relapsing very early (before being eligible for adjuvant chemotherapy, as prior chemotherapy was an exclusion criterion). The patient’s self-assessment of his overall health over the previous seven days seems to be more efficient than subjective evaluations by the physician’s use of the ECOG PS. Indeed, ECOG PS is an evaluation whose reproducibility depends on the assessor. The impact of the ECOG PS on the quality of life and survival of patients is recognized [3,18,19], although some studies have not found a link [11,20]. Question 9 of the QLQ-C30 assesses pain over the previous seven days without specifying the intensity or characteristics of pain, which may explain why we did not find a link between this criterion and tumor progression. We highlight that listening to patients, through simple questions such as QLQ-C30 question 29, could signal tumor progression. Thus, proposing supportive care earlier on the basis of an algorithm based on, among other things, patient-reported outcomes (PROs) could improve patients’ overall survival [12]. Indeed, early palliative care support improved the quality of life and median survival for patients with metastatic lung cancer (11.6 months vs. 8.9 months) [12]. Therefore, this approach has become central to the care of patients in oncology. To optimize the evaluation of the quality of life, computers and e-health tools are being developed. A randomized phase 3 study demonstrated an improvement in the overall survival of patients with metastatic lung cancer when they received follow-up with a web-mediated follow-up algorithm after completing chemotherapy, radiotherapy, or surgery [13], with a 67% reduced risk of death. Follow-up via a mobile application allowed for earlier and, therefore, optimized patient management, since the relapse rate was similar in the two groups.

Regarding the biological data, we observed that tumor markers (CA19-9 and CEA) and their variations during treatment were related to the diagnosis of tumor progression. In our study, we revealed that an increase in CA19-9 levels after diagnosis, measured during tumor assessments, is strongly related to radiological progression, which corroborates the predictive nature of this biological marker for chemotherapy response [21] and makes it a relevant follow-up marker. Although an increase in CA19-9 levels of ≥30 times the standard value at tumor assessments was related to tumor progression in bivariate-adjusted analysis in our study, we were unable to confirm this in multivariate analysis. This shows the potential limitation of using traditional biological markers, for which the cutoff levels vary significantly without a systematic clinical correlation. Regarding its predictive value for response to chemotherapy, CA19-9 correlates with a radiological response with good sensitivity (93.3%) but low specificity (53.2%) [9]. Specifically, a decrease of more than 25% in CA19-9 over the first two cycles of chemotherapy in first-line treatment predicts improved progression-free survival and overall survival [21]. CEA is a tumor marker found in many cancers of the digestive system, including colon cancer, so it is not specific to pancreatic cancer. In our study, we observed that the CEA elevation ≥ threefold the standard value associated or not with an increase in the CEA level was associated with tumor progression in the diagnostic model. This corroborates the studies found in the literature that affirm the interest in this indicator when used in combination with other markers. Indeed, several studies acknowledge that it is of prognostic interest in mPC when combined with other markers, such as CA19-9 [22,23,24]. There is no cutoff CA19-9 definition to specify its impact on the overall survival in pancreatic cancer. Thus, we confirmed that it is a marker of interest for the monitoring of mPC during first-line treatment. However, its lack of specificity led us to believe that its use must be combined with relevant clinical data, such as PRO data.

Regarding inflammation markers, the database did not include all the biological parameters required to calculate the neutrophil-to-lymphocyte, platelet-to-lymphocyte, and lymphocyte-to-monocyte ratios. These ratios, which correlate with a systemic inflammatory response, are recognized as important prognostic factors in many cancers [25,26,27,28,29,30], specifically in pancreatic cancer [19,20,31,32]. These markers should be analyzed and systematically collected when developing a follow-up algorithm in patients with pancreatic cancer during first-line treatment. More recently, liquid biopsy has shown promising results, gaining researchers’ interest in various cancers including PDAC, detection, and tumor heterogeneity assessment [33]. Cell-free DNA (cfDNA) has been identified as a more sensitive and specific marker than CA 19-9 or CEA; thus, the analysis of iterative blood samples might help to better identify patients with progression [34].

This is the first step in the development of an algorithm for the follow-up of those patients with mPC undergoing first-line chemotherapy. The development of a clinical–biological prognostic model could be integrated with a health and personalized medicine approach linked to PRO systems, some of which are used in current practice [35]. The use of connected systems that allow for interaction between the patient and his or her doctor in real life has been studied and is possible [36]. Currently, there is no recommended and reliable monitoring method, and these systems require complex logistics and large-scale development [37]. Denis et al. showed that the use of a smartphone application was possible in those patients undergoing follow-up after lung cancer treatment and was associated with improved survival [38]. Our study legitimizes the development of a similar algorithm to be used in the follow-up of patients with mPC during first-line treatment. An observational pilot study is required to specify the cutoffs that best predict tumor progression and thus define the warning signals that would prompt adaptation of CT scan evaluations. This more dynamic follow-up of patients with mPC during first-line therapy through e-health tools could thus make it possible to detect tumor progression earlier than with standard monitoring with CT scans. Patients would benefit from optimized treatment because, with an early diagnosis of progression, patients are generally in better general condition to benefit from and respond to the second line of chemotherapy. In the future, the connected e-health tools based on this algorithm that incorporate PROs could decrease the number of scans required, thus reducing their emotional and financial burden.

## 5. Conclusions

This exploratory study established the initial steps toward developing an algorithm to diagnose the progression of mPC during first-line therapy. Our study reinforces the idea that measuring PROs alongside biological markers is a good additional method for assessing tumor progression. Such an algorithm could allow patients to be monitored during treatment in a dynamic and personalized way. Based on PROs, through listening to the patient, and by monitoring tumor markers, we could diagnose tumor progression earlier and thus propose new treatments more quickly and reduce the treatment toxicity of ineffective chemotherapy. Conversely, responding patients could continue treatment with clinical and biological follow-up. This study identified the indicators that could validate a clinical–biological algorithm to diagnose tumor progression.

## Figures and Tables

**Figure 1 cancers-14-05068-f001:**
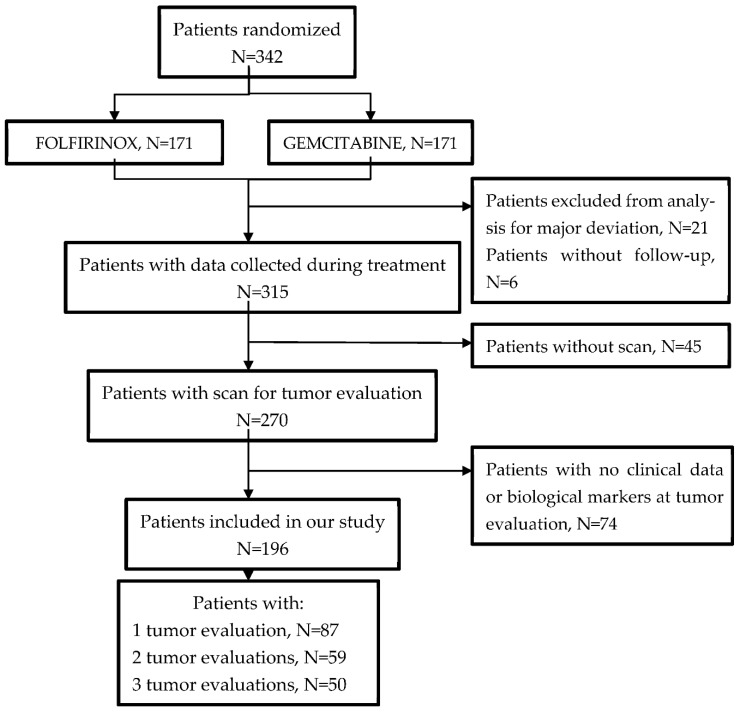
Flowchart: patients from PRODIGE4/ACCORD11 were evaluated for a clinical–biological model of diagnosis of tumor progression for patients with metastatic pancreatic cancer in first-line treatment.

**Table 1 cancers-14-05068-t001:** Patient characteristics at inclusion.

	Total (*n* = 196)	Missing
Chemotherapy arm		-
FOLFIRINOX	100 (51.0)	
Gemcitabine	96 (49.0)	
Sex		-
Women	82 (41.8)	
Men	114 (58.2)	
Age (years)		-
Mean ± STD	58.8 ± 8.8	
≥65	53 (27)	
ECOG performance status		-
0	75 (38.3)	
1	120 (61.2)	
2	1 (0.5)	
Weight at inclusion (kg)		-
Mean ± STD	68.1 ± 13.4	
Body mass index (kg/m^2^)		-
<25	121 (61.7)	
≥25	75 (38.3)	
Initial location		-
Head	74 (37.8)	
Body	66 (33.7)	
Tail	52 (26.5)	
Multi-site	4 (2.0)	
Metastasis location		-
Liver	172 (87.8)	
Peritoneal	33 (16.8)	
Lung	53 (27.0)	
Other	22 (11.2)	
CA19-9 Normalized to standard ^◊^		6
Median (interquartile range)	34.7 (4.9; 388.9)	
Higher than standard value	167 (87.9)	
CEA Normalized to standard ^◊^		4
Median (interquartile range)	2.8 (0.6; 11.3)	
Higher than standard value	125 (65.1)	
Albumin (g/L)		34
Higher than standard value	46 (28.9)	

STD: standard deviation; CA19-9: carbohydrate antigen 19-9; CEA: carcinoembryonic antigen; ^◊^ normalized to the standard of each laboratory. Results are given as frequency and percentage unless otherwise specified.

**Table 2 cancers-14-05068-t002:** Clinical, biological, and toxicity characteristics for the 355 tumor evaluations regardless of their timing.

	Missing	All Evaluations (*n* = 355)
Tumoral progression		
N (%)		92 (25.9)
ECOG performance status	68	
≥2		47 (16.4)
Weight loss since inclusion	-	
≥5%		133 (37.5)
Body mass index (kg/m^2^)	-	
≥25		114 (32.1)
CA19-9 ^‡^		
Higher than standard value	1	278 (78.5)
≥30 times the standard value	-	154 (43.4)
Increased since inclusion	15	86 (25.3)
CEA ^‡^		
Higher than standard value	1	202 (57.1)
≥3 times the standard value	11	104 (30.2)
Increased since inclusion	5	138 (39.4)
Total bilirubin (µmol/L) ^‡^		
Higher than standard value	34	30 (9.4)
AST/SGOT (UI/L) ^‡^		
Higher than standard value	30	143 (44)
ALT/SGPT (UI/L) ^‡^		
Higher than standard value	31	137 (42.3)
GGT (UI/L) ^‡^		
Higher than standard value	49	254 (83)
Quality of life question ^¤^		
“During the past week, have you had pain?”	4	
Not at all/A little		285 (81.2)
Quite a bit/Very much		66 (18.8)
“How would you rate your overall health during the past week?”, 1 to 7 ^¥^	13	
≤4		172 (48.5)
>4		183 (51.5)
Hematological toxicity ^◊^	2	
Grade 2		131 (37)
Grade ≥ 3		84 (23.7)
Other toxicity ^∫^	2	
Grade 2		101 (28.5)
Grade ≥ 3		60 (17)
All toxicity	2	
Grade 2		147 (41.5)
Grade ≥ 3		128 (36.2)

ECOG: Eastern Cooperative Oncology Group; CA19-9: carbohydrate antigen 19-9; CEA: carcinoembryonic antigen; AST/SGOT: aspartate transaminase/serum glutamate oxaloacetic transaminase; ALT/SGPT: alanine transaminase/serum glutamate pyruvate transaminase; GGT: glutamate glutamyl transferase. Results are given as frequency and percentage. * IQR: interquartile range; ^¤^ from the European Organization for Research and Treatment of Cancer, QLQ-C30 version 3; ^¥^ patients had to attribute a number between 1 (very poor) and 7 (excellent) to their overall health over the previous week; ^‡^ normalized to the standard of each laboratory; ^◊^ occurrence of at least one grade 2 or 3 hematological toxicity requiring a reduction in treatment dose during the 2 months before tumor evaluation. The maximal toxicity during the two months before tumor evaluation was considered; ^∫^ occurrence of at least one grade 2 or 3 toxicity except for hematological toxicity during the two months before tumor evaluation. The maximal toxicity during the 2 months before tumor evaluation was considered.

**Table 3 cancers-14-05068-t003:** Clinical and biological factors for diagnosis of tumor progression based on the 355 tumor evaluations: bivariate-adjusted analyses **^§^**.

	OR and Confidence Interval 95%	*p*-Value
Factor at inclusion		
Sex		
Men	1	
Women	0.92 [0.55; 1.53]	0.744
Age (years)		
<65	1	
≥65	0.96 [0.54; 1.69]	0.884
Pancreatic tumor location		
Head	1	
Body	1.22 [0.67; 2.23]	0.507
Tail	0.88 [0.15; 5.09]	0.887
Multi-site	0.97 [0.51; 1.85]	0.919
Metastasis location		
Liver	1.72 [0.77; 3.83]	0.184
Peritoneal	0.73 [0.36; 1.50]	0.388
Lung	0.80 [0.46; 1.42]	0.451
Body mass index (kg/m^2^)		
<25	1	
≥25	0.66 [0.39; 1.13]	0.130
Albumin (g/L) ^‡^		
Higher than standard value	1.36 [0.73; 2.51]	0.329
Factor at tumor evaluation		
ECOG performance status		
0–1	1	
≥2	3.22 [1.58; 6.65]	0.003
Weight loss since inclusion, (kg)		
≥5	1	
<5	1.42 [0.82; 2.47]	0.195
Body mass index (kg/m^2^)		
<25		
≥25	1.83 [0.95; 3.55]	0.068
CA19-9 ^‡^		
Higher than standard value	1.53 [0.74; 3.18]	0.222
≥30 times the standard value	2.85 [1.63; 4.97]	<0.001
Increase since inclusion	3.17 [1.67; 6.02]	0.003
CEA ^‡^		
Higher than standard value	3.09 [1.69; 5.63]	<0.001
≥3 times the standard value	3.80 [2.12; 6.80]	<0.001
Increase since inclusion	2.73 [1.56; 4.78]	<0.001
Total bilirubin (µmol/L) ^‡^		
Higher than normal value	1.69 [0.67; 4.29]	0.234
AST/SGOT (UI/L) ^‡^		
Higher than standard value	1.21 [0.69; 2.12]	0.489
ALT/SGPT (UI/L) ^‡^		
Higher than standard value	1.12 [0.63; 1.99]	0.688
GGT (UI/L) ^‡^		
Higher than standard value	2.87 [1.05; 7.87]	0.042
Hematological toxicity ^◊^		
Grade 2	1.31 [0.74; 2.33]	0.353
Grade ≥ 3	0.79 [0.39; 1.61]	0.508
Other toxicity ^∫^		
Grade 2	1.21 [0.66; 2.22]	0.523
Grade ≥ 3	2.08 [1.07; 4.05]	0.032
All toxicity		
Grade 2	0.91 [0.46; 1.79]	0.783
Grade ≥ 3	1.10 [0.56; 2.16]	0.780

ECOG: Eastern Cooperative Oncology Group; CA19-9: carbohydrate antigen 19-9; CEA: carcinoembryonic antigen; AST/SGOT: aspartate transaminase/serum glutamate oxaloacetic transaminase; ALT/SGPT: alanine transaminase/serum glutamate pyruvate transaminase; GGT: glutamate glyoxylate aminotransferase; ^§^ adjusted on treatment arms; ^‡^ normalized to the standard of each laboratory; ^◊^ occurrence of at least one grade 2 or 3 hematological toxicity requiring a reduction in treatment dose during the two months before tumor evaluation. The maximal toxicity during the two months before tumor evaluation was considered; ^∫^ occurrence of at least one grade 2 or 3 toxicity except for hematological toxicity during the two months before tumor evaluation. The maximal toxicity during the two months before tumor evaluation was considered.

**Table 4 cancers-14-05068-t004:** Quality of life score by threshold.

Quality of Life Score by Threshold, ≤33; [33 à 66]; ≥67 ^§^	N (%)	OR and Confidence Interval 95%	*p*-Value
Physical functioning			
≤66	97 (27.4%)	1.37 [0.77; 2.44]	0.266
≥67	257 (72.6%)	1	
Role functioning			
≤66	194 (55.1%)	1.42 [0.83; 2.44]	0.190
≥67	158 (44.9%)	1	
Emotional functioning			
≤66	109 (31.5%)	1.32 [0.76; 2.3]	0.319
≥67	237 (68.5%)	1	
Cognitive functioning			
≤66	88 (25.3%)	2.09 [1.16; 3.77]	0.017
≥67	260 (74.7%)	1	
Social functioning			
≤66	158 (45.9%)	1.57 [0.92; 2.68]	0.093
≥67	186 (54.1%)	1	
Global health status/Quality of life			
≤33	57 (16.6%)	3.41 [1.37; 8.48]	0.010
34 to 66	203 (59.2%)	2.53 [1.17; 5.48]	0.020
≥67	83 (24.2%)	1	
Fatigue			
≤33	307 (87.0%)	1	
≥34	46 (13.0%)	1.79 [0.84; 3.82]	0.123
Nausea and vomiting			
≤33	308 (87%)	1	
≥34	46 (13%)	1.52 [0.71; 3.23]	0.261
Pain			
≤33	280 (79.1%)	1	
≥34	74 (20.9%)	1.36 [0.73; 2.51]	0.316
Dyspnoea			
≤33	319 (90.4%)	1	
≥34	34 (9.6%)	0.76 [0.29; 1.99]	0.550
Insomnia			
≤33	304 (86.4%)	1	
≥34	48 (13.6%)	1.36 [0.64; 2.89]	0.400
Appetite loss			
≤33	255 (72.2%)	1	
≥34	98 (27.8%)	1.87 [1.07; 3.26]	0.030
Constipation			
≤33	300 (86.2%)	1	
≥34	48 (13.8%)	1.09 [0.51; 2.32]	0.822
Diarrhea			
≤33	302 (87%)	1	
≥34	45 (13%)	0.57 [0.22; 1.46]	0.230
Financial difficulties			
≤33	329 (95.4%)	1	
≥34	16 (4.6%)	1.62 [0.34; 7.78]	0.445

^§^ Thresholds defined in the quality of life study of PRODIGE4/ACCORD11.

**Table 5 cancers-14-05068-t005:** Clinical and biological factors for diagnosis of tumor progression based on the 355 tumor evaluations: multivariate analysis.

	OR and Confidence Interval 95%	*p*-Value
Chemotherapy		
FOLFIRINOX	1	
Gemcitabine	3.47 [1.94; 6.18]	<0.001
Global health status/Quality of life ^¤, ø^		
≤33	3.38 [1.15; 9.91]	0.028
34–66	2.65 [1.06; 6.59]	0.037
≥67	1	
CA19-9 at TE > CA19-9_inclusion_ ^◊^	2.63 [1.27; 5.43]	0.014
CEA at TE		<0.001
<3 times CEA laboratory ^‡^	1	
≥3 times CEA laboratory ^‡^ and ≤ CEA inclusion ^◊^	2.46 [1.12; 5.40]	
≥3 times CEA laboratory ^‡^ and > CEA inclusion ^◊^	5.94 [2.51; 14.03]	

CA19-9: carbohydrate antigen; CEA: carcinoembryonic antigen; ^¤^ from European Organization for Research and Treatment of Cancer, QLQ-C30 version 3; ^ø^ data collected during tumor evaluation; ^‡^ standard value of the laboratory; ^◊^ value at inclusion.

## Data Availability

Data will be made available upon request.

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
