# Peer review of "Development of a Clinical–Biological Model to Assess Tumor Progression in Metastatic Pancreatic Cancer: Post Hoc Analysis of the PRODIGE4/ACCORD11 Trial"

_cancers, 2022, doi:10.3390/cancers14205068_

Round 1

Reviewer 1 Report

Overall, this is an interesting manuscript using high quality clinical trial data, albeit it is a retrospective analysis. Nonetheless, the authors hypothesis that radiographic surveillance alone can be improved is clinically sound. The data they present is intriguing and clinically relevant to practice. The statistical analysis performed is sound and appropriate in my opinion. I agree with the authors statement in the discussion that the "discriminating power of our tumor progression diagnosis model is nevertheless too weak to confirm the progression diagnosis with certainty", the data supports the investigators overall hypothesis which is intriguing and further studies to validate the findings would be of worthy of pursuit. I think the article will be of broad interest to clinicians caring for PDAC patients. I recommend the article be accepted for publication. 

Author Response

Review

Answer

Modification in text

#1

Overall, this is an interesting manuscript using high quality clinical trial data, albeit it is a retrospective analysis. Nonetheless, the authors hypothesis that radiographic surveillance alone can be improved is clinically sound. The data they present is intriguing and clinically relevant to practice. The statistical analysis performed is sound and appropriate in my opinion. I agree with the authors statement in the discussion that the "discriminating power of our tumor progression diagnosis model is nevertheless too weak to confirm the progression diagnosis with certainty", the data supports the investigators overall hypothesis which is intriguing and further studies to validate the findings would be of worthy of pursuit. I think the article will be of broad interest to clinicians caring for PDAC patients. I recommend the article be accepted for publication.

Thank you for your supportive review, it’s a pleasure to have our main message fully understood; we nevertheless corrected some mistakes and typos in English.

See full manuscript with edited text.

Reviewer 2 Report

This is a well designed and conducted retrospective analysis of a phase III clinical data aiming in the development of a predictive algorithm in metastatic pancreatic cancer. 

Althought out of the scope of its analysis, a comment regarding the number of patients having previous surgery (eg pancreatectomy) and if they had diffent quality of life outcomes would be of interest.

This paper can be accepted in its present form for publication.

Author Response

Review

Answer

Modification in text

#2

This is a well designed and conducted retrospective analysis of a phase III clinical data aiming in the development of a predictive algorithm in metastatic pancreatic cancer. 

Althought out of the scope of its analysis, a comment regarding the number of patients having previous surgery (eg pancreatectomy) and if they had diffent quality of life outcomes would be of interest.

This paper can be accepted in its present form for publication.

Thank you for this very pertinent comment, we tried to incorporate it as much as possible, as we did not have this data for further analysis.

There were 15 and 10 patients in the FOLFIRINOX and gemcitabine arms, respectively, who underwent surgery before relapsing very early (before being eligible for adjuvant chemotherapy, as prior chemotherapy was an exclusion criterion). We have no data at all on a possible difference in quality of life for these patients, but this raises an interesting question that we have brought to the discussion.

See discussion section, paragraph 2:

It should be noted that we do not have specific quality of life data on patients who have had previous surgery; there were 15 and 10 patients in the FOLFIRINOX and gemcitabine arms, respectively, who underwent surgery before relapsing very early (before being eligible for adjuvant chemotherapy, as prior chemotherapy was an exclusion criterion).